# Multidrug-Resistant Phenotypes of *Escherichia coli* Isolates in Wild Canarian Egyptian Vultures (*Neophron percnopterus majorensis*)

**DOI:** 10.3390/ani11061692

**Published:** 2021-06-06

**Authors:** Alejandro Suárez-Pérez, Juan Alberto Corbera, Margarita González-Martín, María Teresa Tejedor-Junco

**Affiliations:** 1Wildlife Animal Rescue Centre, Cabildo de Tenerife, 38291 La Laguna, Spain; alejandro.suarezperez@ulpgc.es; 2Department of Animal Pathology, Animal Production and Food Hygiene and Technology, University of Las Palmas de Gran Canaria (ULPGC), 35413 Arucas, Spain; 3Research Institute of Biomedical and Health Sciences, ULPGC, 35016 Las Palmas, Spain; margaritarosa.gonzalez@ulpgc.es (M.G.-M.); mariateresa.tejedor@ulpgc.es (M.T.T.-J.); 4Department of Clinical Sciences, University of Las Palmas de Gran Canaria (ULPGC), 35413 Arucas, Spain

**Keywords:** multidrug resistance, antimicrobial resistance, *E. coli*, Canarian Egyptian vultures, wildlife

## Abstract

**Simple Summary:**

Increasing antimicrobial resistance is a global problem for both human and animal health. *Escherichia coli* is frequently used as a “sentinel” for antimicrobial resistance and as an indicator of faecal contamination of the environment. This study is a characterisation of the antimicrobial resistance phenotypes of *E. coli* isolates obtained from cloacal samples of Canarian Egyptian vultures. A total of 65 chickens and 38 adult and immature birds were studied. Antimicrobial susceptibility to 16 antibiotics of 12 different categories was determined in 103 *E. coli* isolates. We found a 39.8% prevalence of multidrug-resistant (MDR) *E. coli*. Almost all MDR phenotypes found included resistance to tetracycline, an antibiotic widely used in veterinary medicine. Resistance has also been found to chloramphenicol (13 MDR phenotypes), imipenem (5 MDR phenotypes) and others. Wild birds can act as reservoirs and disseminators of MDR *E. coli*, transferring them via faeces to the environment, feed or water. Our results highlight the need to minimise exposure of wild birds to antimicrobials from human activities to avoid the spread of antimicrobial resistance.

**Abstract:**

The presence of multidrug-resistant (MDR) *Escherichia coli* in cloacal samples from Canarian Egyptian vultures was investigated. Samples were obtained from chicks (*n* = 65) and from adults and immature birds (*n* = 38). Antimicrobial susceptibility to 16 antibiotics included in 12 different categories was determined for 103 *E. coli* isolates. MDR was defined as acquired non-susceptibility to at least one agent in three or more antimicrobial categories. Forty-seven different resistance phenotypes were detected: 31 MDR (41 isolates) and 16 non-MDR (62 isolates). One isolate was resistant to all 12 antimicrobial categories and 2 phenotypes included resistance to 9 antimicrobial categories. Imipenem resistance was included in five MDR phenotypes, corresponding to five different isolates. Statistically significant differences in prevalence of MDR-phenotypes were found between chicks in nests and the rest of the animals, probably due to the shorter exposure time of chicks to antimicrobials. The main risk derived from MDR bacteria in scavengers is that it threatens the treatment of wild animals in rescue centres and could be transferred to other animals in the facilities. In addition to this, it could pose a health risk to veterinarians or other staff involved in wildlife protection programmes.

## 1. Introduction

The increase of antimicrobial resistance is a global problem for both human and animal health [1,2]. Antimicrobial resistance has been detected in many wildlife species on all continents, including Antarctica and the Arctic [3,4,5,6,7,8,9], even when wildlife are not treated with these compounds. This supports the possibility of transmission of resistant bacteria between human, livestock and domestic animals, wildlife and the environment, as well as the possible selective pressure exerted by antibiotic residues present in the environment [10,11,12,13,14]. Surveillance of antimicrobial resistant (AMR) bacteria in wildlife can help to understand the extent of the problem and propose possible solutions [15].

Several definitions of multidrug-resistant (MDR), extensively drug-resistant (XDR) and pandrug-resistant (PDR) bacteria have been proposed. A joint initiative of the European Centres for Disease Control and Prevention (ECDC) and the USA Centres for Disease Control and Prevention (CDC) created an expert group to standardise the terminology used to describe resistance profiles [16]. After defining the categories of antimicrobials used for the treatment of infections caused by different groups of bacteria, MDR was defined as acquired non-susceptibility to at least one agent in three or more antimicrobial categories, XDR was defined as non-susceptibility to at least one agent in all but two or fewer antimicrobial categories (i.e., bacterial isolates remain susceptible to only one or two categories) and PDR was defined as non-susceptibility to all agents in all antimicrobial categories. For all three definitions, non-susceptibility refers to a resistant, intermediate or non-susceptible result obtained in in vitro antimicrobial susceptibility testing. 

*Escherichia coli* is frequently used as a “sentinel” of antimicrobial resistance [17] and is an important component of the gastrointestinal microbiota of several animal species (including humans) and is also used as an indicator of faecal contamination of the environment [18]. Despite the commensal nature of *E. coli*, this species is an opportunistic pathogen and can be involved in human and animal infections. Moreover, it has a strong ability to transfer these resistance genes in vivo to other Gram-negative bacteria such as pathogenic strains of *E. coli* or *Salmonella* [4,19]. The most commonly recovered microorganism from raptors with clinical signs of septicaemia or respiratory illness is *E. coli* [20].

Antibiotics residues in animal carcasses represent a risk for avian scavengers like vultures, that are actively exposed to antibiotics by ingestion of carrion. In Spain, Blanco et al. [21] demonstrated a correlation between the prevalence of MDR *E. coli* and other species and intensification in animal production. Other publications in our country have also demonstrated the involvement of birds in the spread of antibiotic resistance in the ecosystem [17,22,23].

Wild birds are highly mobile and can act as reservoirs and spreaders of MDR *E. coli*, transferring them via faeces to the environment, animal feed or water [7,11]. The aim of our work was the analysis of MDR-phenotypes in *E. coli* isolated from a bird-of-prey endemic to the Canary Islands (Spain): the Canarian Egyptian vulture (*Neophron percnopterus majorensis*), included in the Spanish Catalogue of Threatened Species under the category “In Danger of Extinction” (Royal Decree 139/2011) [24]. Because the Canarian vulture is a sedentary population, the potential spread of *E. coli* MDR would be limited to the islands they inhabit. In addition, Canarian Egyptian vultures tend to live in pairs rather than flocks, so the risk of environmental contamination with MDR bacteria is probably lower than the described for geese or gulls [3,25], but could pose a health risk to veterinaries, nature conservation workers or ornithologists [26]. The epidemiologically significant antimicrobial categories proposed for *Enterobacteriaceae* were applied to the *E. coli* isolates included in this study [16]. 

The main objective of this study is to determine the prevalence of MDR-*E. coli* in faecal samples of Canarian Egyptian vultures of different ages.

## 2. Material and Methods

Within a long-term monitoring program of Canarian Egyptian vultures (*Neophron percnopterus majorensis*) [27], cloacal samples were obtained from 142 animals: 30 samples in 2015, 62 in 2016 and 50 in 2017. 

All procedures, including the methods of capture and handling of wild vultures, were carried out under the project license approved by The Biodiversity Directorate of the Government of the Canary Islands; the official reference numbers of the committing authority are: 2014/256, 2015/1652 and 2016/1707. 

Adult and immature birds were trapped at baited sites using cannon-netting. Chicks were captured during the fledgling stage in the nests. In the vast majority of cases, there was only one vulture chick in each nest. All animals were apparently healthy. 

Cloacal swabs (81 from chicks in the nest and 61 from wild adults) were obtained, placed in an Amies blue plastic/viscose gel transport medium (Darmstadt, Germany) and stored at 4 °C until arriving at the Microbiology Laboratory within 24 h.

For the detection of *E. coli*, samples were cultured on MacConkey agar (BD Difco, Detroit, MI, USA) and incubated overnight at 37 °C. Colonies with typical *E. coli* characteristics were identified using an automated Vitek^®^ 2 system (bioMérieux, Marcy L’Etoile, France) according to the manufacturer’s instructions.

Antimicrobial susceptibility tests were performed by the disc diffusion method according to CLSI methods (CLSI 2015). Antibiotics tested are shown in Table 1. All antimicrobial resistance phenotypes were tested to determine whether they met the definition of multidrug-resistance.

For comparison, antimicrobial resistance results related to animal age, chi-square with the Yates’s continuity correction were calculated. SPSS v. 22.0 (SPSS Inc., Chicago, IL, USA) was used to compare data sets. Statistical significance was set at *p* < 0.05.

## 3. Results

Forty-seven different resistance phenotypes were observed (Table 2), including 26 isolates that were pan-susceptible. Sixteen phenotypes included resistance to less than three different categories of antimicrobials and were therefore considered non-MDR. The most frequent non-MDR profile was “AM, SXT” (8 isolates). Thirty-one different MDR phenotypes were found, including resistance to 3 to 12 categories of antimicrobials. The most frequent one was “AM, PIP, TE, and SXT” (6 isolates). One isolate was resistant to 12 antimicrobial categories and 2 phenotypes included resistance to 9 antimicrobial categories.

We tested 12 of 17 proposed categories, therefore, we can describe isolates showing an MDR phenotype. Imipenem, a carbapenem antibiotic classified as “critically important for human medicine” [2] and restricted to hospital use, was tested, and five MDR phenotypes included resistance to this antibiotic. Resistance to imipenem was found in five isolates, showing five different MDR phenotypes.

*E. coli* was obtained in samples of 65 chicks in nests, and among them, 18 were MDR and 47 non-MDR. Among the remaining animals (38 isolates), 23 were MDR and 15 non-MDR (Table 2). Overall, of the 103 *E. coli* isolates, 41 (39.8%) were considered MDR and 62 (60.2%) non-MDR.

We found statistically significant differences between chicks and the rest of the animals (χ^2^ = 9.463; *p*-value = 0.002), indicating a higher probability of MDR *E. coli* detection in adults and immature birds than in chicks in nests.

## 4. Discussion

In a previous manuscript [28], the percentages of resistance to different antibiotics in bacteria isolated from vulture cloacal swabs were analysed. The coincidence of resistance to several antibiotics in the same isolate was not studied, so multidrug resistance phenotypes were not described.

In this work, the antimicrobial resistance phenotypes of *E. coli* isolates obtained from cloacal samples of Canarian Egyptian vultures were characterised. The definition of multidrug resistance proposed by Magiorakos et al. [16] was applied. Since not all the antimicrobial categories proposed for Magiorakos et al. [16] were included in our study, we cannot confirm the existence of XDR or PDR isolates. Their proposal includes categories such as glycylcyclines, monobactams or antipseudomonal penicillins + β-lactamase inhibitors, which are not commonly used in veterinary therapy for infections due to *Enterobacteriaceae*. 

Stedt et al. [29] compared the presence of MDR *E. coli* in gulls (*Laridae*) from nine countries, describing a total of 59 MDR phenotypes, a higher result than ours (31 different MDR-phenotypes). In different areas of Spain, they found prevalence ranging from 28.6 to 45.3%, a similar result to the one found in Canarian Egyptian vultures (39.8%). On the other hand, Sharma et al. [30] found a very high incidence of MDR *E. coli* in Egyptian vultures fed on carcass dumps in India. These differences could be due to differences in clinical and veterinary use of antimicrobials in both countries. Atterby et al. [25] described an 83% prevalence of MDR *E. coli* when tested ESBL producing *E. coli* from Swedish gulls.

Almost all MDR phenotypes found included resistance to TE, which could be related to the frequent use of this antibiotic in veterinary medicine [31]. Chloramphenicol was banned for agricultural use in Europe more than 25 years ago, but 13 MDR phenotypes of resistance to this drug were found.

Five phenotypes included resistance to imipenem. The emergence of carbapenem resistant *Enterobacterales* is a global health problem. Carbapenems are used in hospital settings as a last-line treatment for severe human infections. In veterinary tertiary care centres, carbapenem is used in selected clinical cases in companion animals [32]. Carbapemenase-producing bacteria have been previously described in wild and domestic animals [32,33,34,35,36,37,38]. It appears that carbapenem-resistant bacteria or carbapenem residues are making their way out of hospitals and environmental reservoirs could be created.

We found a statistically lower probability of MDR-*E. coli* isolation in chicks in nests compared to all other animals. As we have previously proposed [28], the differences could be due to the fact that chicks in nests have been exposed to resistant bacteria or environmental antimicrobial residues for less time than adult and immature birds. 

Many authors suggest that antimicrobial resistance in wildlife bacteria is closely related to resistance in humans and domestic animal strains [10,25,39], but multidrug-resistant bacteria have also been detected in wildlife in remote areas with little if no human contact [5,9]. 

The likelihood of MDR bacteria being transmitted from wild birds to humans and domestic animals depends on several factors: the rate of colonisation, the intensity of faecal shedding, the survival of MDR bacteria in the environment, the infective dose and the ability to infect human and domestic animals [40]. Faeces from wild birds are shed directly into the environment, potentially exposing human and animal populations to MDR bacteria [41]. Some outbreaks of human or livestock infections associated with the shedding pathogen bacteria by wild birds have been described [42,43]. 

Not all wild birds pose the same risk for public health, but waterfowl flocks can easily contaminate humans and domestic animal water supplies. In our opinion, scavengers are less likely to do the same. 

As suggested by the VKM report in 2018 [44], the risk of transmission of MDR bacteria cannot be estimated either quantitatively or qualitatively with the information available at this time.

In our opinion, the main risk arising from MDR-bacteria in scavengers is that it threatened the treatment of wild animals in rescue centres and could be transferred to other animals in the facilities. In addition, it could pose a health risk to veterinarians or other personnel involved in wildlife protection programmes. We consider that scavengers are a reflection of anthropogenic antimicrobial contamination in the environment and also could acquire MDR bacteria from animal or human sources, but the drivers of MDR resistance are likely to be more complex than just anthropogenic causes [45]. As suggested by Blanco and Bautista [46], supplementary feed for scavengers should be sourced from farms that guarantee the absence of antibiotic residues in the feed.

## 5. Conclusions

The results of our study provides evidence for the presence of MDR-bacteria in scavengers. Different categories of phenotypes have been found in the chicks in nests and in immature and adult animals. The 39.8% of *E. coli* isolates were considered MDR. A higher prevalence of MDR *E. coli* isolates was found in adults and immature birds compared to chicks’ *E. coli* isolates.

Almost all MDR phenotypes found included resistance to tetracycline, an antibiotic widely used in veterinary medicine. Resistance was also found to chloramphenicol (13 MDR phenotypes), imipenem (5 MDR phenotypes) and other clinically important antibiotics.

Wild birds can act as reservoirs and disseminators of MDR *E. coli*, transferring them via faeces to the environment, feed or water. 

Our results highlight the need to minimise the exposure of wild birds to antimicrobials from human activities to avoid the spread of antimicrobial resistance. The management of this problem requires a “One Health” approach.

## Figures and Tables

**Table 1 animals-11-01692-t001:** Antimicrobial categories and antimicrobial agents tested for defining MDR in *E. coli* isolates (adapted from Magiorakos et al. 2012).

Antimicrobial Categories (Code)	Antimicrobial Agents	Abbreviation and Charge of Disks
Aminoglycosides (A)	Amikacin Gentamicin Tobramycin	AK (30 µg) GM (30 µg) NN (10 µg)
Carbapenems (Ca)	Imipenem	IPM (10 µg)
Non-extended spectrum cephalosporins: 1st and 2nd generation cephalosporins (1–2 Ce)	Cephalexin	CEP (30 µg)
Extended-spectrum cephalosporins: 3rd and 4th generation cephalosporins (3–4 Ce)	Cefpodoxime	CPD (10 µg)
Fluoroquinolones (Fl)	Enrofloxacin Marbofloxacin	ENO (5 µg) MAR (5 µg)
Folate pathway inhibitors (Fo)	Trimethoprim/ Sulfamethoxazole	SXT (1.25 µg + 23.75 µg)
Penicillins (Pe)	Ampicillin Piperacillin	AM (10 µg) PIP (100 µg)
Penicillins + β-lactamase inhibitors (Pβ)	Amoxicillin/ Clavulanic Acid	AMC (20 µg + 10 µg),
Phenicols (Ph)	Chloramphenicol	C (30 µg)
Polymyxins (Po)	Polymyxin B	PB (300 U)
Tetracyclines (T)	Tetracycline	TE (30 µg)
Nitrofuranes (N)	Nitrofurantoin	F/M (300 µg)

**Table 2 animals-11-01692-t002:** Antimicrobial resistance phenotypes. The antimicrobials have been ordered by their frequency in the isolates and the number of antimicrobial categories. The number of isolates in each group of age is also included.

CAT.	Fo	Pe	T	Pe	Fl	Fl	Ph	1–2 Ce	Pβ	A	N	A	Ca	3–4 Ce	Po	A	No. of CAT.	No. of Isolates	C *		A *
ATB.	SXT	AM	TE	PIP	ENO	MAR	C	CEP	AMC	GM	F/M	NN	IPM	CPD	PB	AK					
**non-MDR**		AM															1	3	2		1
		AM		PIP													1	1			1
							C										1	7	5		2
					ENO	MAR											1	1	1		
	SXT																1	1	1		
			TE														1	3	3		
		AM			ENO												2	1	1		
		AM			ENO	MAR											2	2	1		1
	SXT	AM		PIP													2	2	2		
	SXT	AM															2	8	8		
	SXT				ENO	MAR											2	1	1		
			TE		ENO	MAR											2	1	1		
	SXT									GM							2	1	1		
	SXT														PB		2	1	1		
			TE				C										2	3	1		2
	SXT		TE														2	2	1		1
	SXT	AM	TE	PIP													3	6	2		4
	SXT	AM	TE														3	1	1		
	SXT		TE		ENO	MAR											3	1	1		
	SXT	AM					C										3	1			1
	SXT	AM					C										3	1			1
**MDR**	SXT	AM	TE	PIP	ENO	MAR			AMC								4	1	1		
		AM	TE				C	CEP									4	1			1
	SXT	AM	TE		ENO	MAR											4	1			1
	SXT	AM	TE	PIP				CEP									4	1			1
	SXT	AM	TE	PIP	ENO	MAR	C				F/M						4	1	1		
	SXT	AM	TE	PIP	ENO	MAR											4	1	1		
	SXT	AM	TE	PIP	ENO	MAR											4	2			2
	SXT	AM	TE	PIP	ENO												4	1	1		
	SXT	AM	TE	PIP						GM		NN					4	1			1
	SXT	AM	TE	PIP			C										4	3	2		1
	SXT	AM	TE	PIP							F/M						4	1			1
	SXT	AM	TE				C										4	2	1		1
	SXT		TE				C			GM							4	1	1		
	SXT	AM	TE	PIP			C	CEP									5	1			1
	SXT	AM	TE	PIP	ENO		C										5	1			1
	SXT	AM	TE	PIP	ENO	MAR						NN					5	1	1		
	SXT	AM	TE	PIP				CEP	AMC				IPM				6	1			1
	SXT	AM	TE		ENO	MAR	C			GM	F/M						6	1	1		
	SXT	AM	TE	PIP	ENO	MAR	C			GM		NN					6	1	1		
	SXT	AM	TE		ENO	MAR	C	CEP	AMC								7	1	1		
	SXT	AM	TE	PIP				CEP	AMC				IPM	CPD			7	1			1
	SXT	AM	TE	PIP	ENO	MAR	C		AMC		F/M						7	1			1
	SXT	AM	TE	PIP	ENO	MAR	C	CEP	AMC								9	1	1		
	SXT	AM	TE	PIP	ENO			CEP	AMC		F/M		IPM	CPD			9	1			1
	SXT	AM	TE	PIP	ENO	MAR	C	CEP	AMC	GM	F/M	NN	IPM	CPD	PB	AK	12	1			1
**SUCEPTIBLE TO ALL ANTIMICROBIALS TESTED**																		12	26	18	8

* Distribution of the isolate in the different groups of age (C = Chicks in the nests; A = Adults and immature birds). The abbreviations for families and antibiotics used in Table 2 are the same as those used in Table 1. The color as used for a better visualization of the different phenotypes.

## Data Availability

Data sharing not applicable.

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
