# Peer review of "Multidrug-Resistant Phenotypes of Escherichia coli Isolates in Wild Canarian Egyptian Vultures (Neophron percnopterus majorensis)"

_animals, 2021, doi:10.3390/ani11061692_

Round 1
Reviewer 1 Report
In terms of methodology, the manuscript is correctly written, but there are some doubts about the reuse of data (previously described in reference 25). In the current version, the authors have not properly explained clearly what the innovation of this work is compared to the previous one. In fact, some information has been reused (lines 166-172).
Unfortunately, the short statement (lines 143-146) is not sufficient to exclude that this is a duplication of a previous publication. The authors should clearly indicate what is different in the present work and justify the need to discussion the previously obtained and described results, which additionally included only Salmonella and other Enterobacteriaceae strains (a total of 25 additional strains)
Minor comments
Line 21: what does "other important antibiotics" mean?
Line 96: Were cloaca swabs taken from one young from each nest or from more birds from the same nest? (this may affect the similarity of the resistance profiles via the same source)
Table 2: the drug abbreviations should be included in the legend
Line 200: better sounds: "As suggested (name of author (s) [43]"
Author Response
Please see the attachment.
Question in Blue
Answer in Yellow.

Reviewer 2 Report
Multidrug-resistant phenotypes of Escherichia coli isolates in wild Canarian Egyptian vultures (Neophron percnopterus majorensis) is a well-designed, well-written work and it is appreciated that they work in such a challenging environment, but so necessary to understand the epidemiology of bacteria and its MDRs in the environment. However, there are a few points that must be corrected before they can be published.
In the first place, a clear objective must be included in the main manuscript to define the conclusions obtained from the study, in the same way, they are contemplated in the abstract and the simple summary.
Check if the Canarian Egyptian vultures is classified as a raptor or scavenger in the bibliography. In the manuscript, all are cited as a raptor. Review this point.
In the introduction section, much has been said about the nomenclature or classification of resistance and the studies carried out on multi-resistance in wild birds worldwide. However, little has been said about results obtained in populations of the same country in recent years, supporting the results obtained in this work. Update the introduction.
Lines 143-144, replace “…,we analysed percentages of resistance to different antibiotics in bacteria isolated from vulture cloacal swabs” by “…, the percentages of resistance to different antibiotics in bacteria isolated from vulture cloacal swabs were analyzed”.
Lines 147-148, replace “…, we characterised the antimicrobial resistance phenotypes of E. coli isolates obtained from cloacal samples of Canarian Egyptian vultures” by “…, the antimicrobial resistance phenotypes of E. coli isolates obtained from cloacal samples of Canarian Egyptian vultures were characterized”
Lines 131-132, Include the standard error with the values in percentage of E. coli that have been obtained in the statistical analysis.
Line 119, include the P-value obtained at the end of the sentence “The most frequent non-MDR profile was “AM, SXT” (8 isolates)”.
Line 121, include the P-value obtained at the end of the sentence “The most frequent onewas “AM, PIP, TE, SXT” (6 isolates)”.
Include in discussion what the relevant result obtained may be due to, such as that "indicating a higher probability of MDR E. coli detection in adults 134 and immature birds than in chicks in nests"
Author Response
Please see the attachment.
Question in Blue
Answer in yellow

Round 2
Reviewer 1 Report
I do not have any other suggestions